# Dynamic Pattern and Evolution Trend of the New Four Modernizations Synchronous Development in China: An Analysis Based on Panel Data from 31 Provinces

Yang Li [1,2,3], Kunlin Zhu [1,2,*], Xianghui Li [1,2], Zunirah Mohd Talib [3] and Brian Teo Sheng Xian [3,*]

[1] School of Science, Henan University of Technology, Zhengzhou 450001, China; 012021090918@gsm.msu.edu.my (Y.L.)
[2] School of Economics and Trade, Henan University of Technology, Zhengzhou 450001, China
[3] Graduate School of Management, Management and Science University, Shah Alam 40100, Malaysia
* Correspondence: hautzkl@126.com (K.Z.); brian_teo@msu.edu.my (B.T.S.X.)

**Abstract:** The New Four Modernizations (NFM) synchronous development strategy proposed by the Chinese government is an important part of Chinese modernization and is of great significance in promoting the sustainable and high-quality development of the social economy. This paper aims to provide a framework for analyzing the dynamic evolution pattern of China's NFM synchronous development. We expounded on the theoretical basis of China's NFM synchronous development strategy. Then, the entropy method and convergence model were used to analyze the dynamic pattern and evolution trend of the NFM synchronous development in China. The findings are as follows: (1) In terms of the current situation of NFM development, China's new urbanization and industrialization were in the middle stage, while informationization and agricultural modernization were in the initial stage of rapid development. (2) The level of China's NFM synchronous development continued to improve, but the problem of unbalanced, uncoordinated, and unsynchronized development remained prominent. There were great differences in the development levels of China's new urbanization, industrialization, informationization, and agricultural modernization, which showed that there was a problem of unbalanced and unsynchronized development within China's NFM. On the other hand, the synchronous development level of NFM in the eastern, central, and western regions of China showed a spatial pattern of decline although the eastern region was clearly higher than the central and western regions; this revealed that the regional development of NFM in China was unbalanced and uncoordinated. (3) China's NFM synchronous development has significant $\sigma$-convergence and $\beta$-convergence, indicating that backward regions are catching up and regional differences will gradually narrow in the future. This study is helpful in understanding the current situation as well as the dynamic pattern and future evolution trend of NFM synchronous development in emerging countries such as China.

**Keywords:** New Four Modernizations; synchronous development; entropy method; convergence model; dynamic pattern; evolution trend



## 1. Introduction

At the beginning of reform and opening up, due to limitations in human, material, financial and technological levels, China adopted an unbalanced development strategy [1] which played a very positive role in its social and economic development during a specific period. At present, China has become the second largest economy in the world, with its overall national strength and international status significantly enhanced [2]. However, with the continuous advancement of China's modernization process, the problems brought about by the unbalanced development strategy, such as unbalanced regional development, uncoordinated urban and rural development, and environmental pollution, have become prominent [3–6]. In this context, the Chinese government proposed to constantly enhance

the integrity of development and promote the synchronous development of new urbanization, industrialization, informatization, and agricultural modernization, [7] also known as the New Four Modernizations (NFM) synchronous development strategy.

New urbanization, industrialization, informatization, and agricultural modernization are important aspects of China's modernization construction [8]. NFM synchronous development is an important strategic task of the Chinese government in a new round of reform, and an inevitable choice to promote the sustained, stable, and healthy development of the social economy [9]. In terms of urbanization, China's urban functions and living environments have continuously improved [10,11]. China's urbanization rate reached 64.72% in 2021; urban gas penetration rate was 98.0%; water supply rate was 99.4%; and the green space rate in urban built-up areas was 38.7%. However, land urbanization is faster than population urbanization, the spatial structure is not reasonable, and urban management levels are not high. In terms of industrialization, China has built the most complete modern industrial system worldwide and has become the world's largest manufacturing country [12]. However, China's industrialization process is characterized by high investment, high output, high energy consumption, and high pollution [13–15]. There is an urgent need for a new industrialization road with high scientific and technological content, good economic benefits, low resource consumption, low environmental pollution, and full utilization of human resource advantages. In recent years, a new round of scientific and technological revolution supported by information technology has emerged [16]. New technologies, industries, and business forms such as big data, artificial intelligence, the digital economy, and e-commerce continue to emerge [17–20]. Information network technology is becoming the main driving force for economic development [21–24]. In 2021, the number of internet users in China reached 1.032 billion, the internet penetration rate reached 73%, and the scale of mobile payments in China ranked first in the world. China has built the world's largest and most advanced network infrastructure, and its information industry is growing rapidly, contributing more to economic development. However, there is still a large gap between China's informatization and developed countries; the country has a weak ability for independent innovation, low levels of application, and prominent information security problems [24,25]. In terms of agricultural modernization, agricultural science and technology have made continuous progress, the rural economy has developed rapidly, and farmers' living standards have continuously improved [26–28]. However, the production mechanism is backward, the operation is decentralized, the quality of employees is not high, and agricultural equipment is insufficient [29–31].

New urbanization, industrialization, informatization, and agricultural modernization are not isolated, but an organic whole that is interrelated and mutually reinforcing. The integration of new urbanization and information technology can help promote the equalization of basic public services through the implementation of "internet +" [32–34]. The deep integration of new industrialization and informatization can give birth to the emergence of new business models and promote the clustering and ecological development of strategic emerging industries [35–37]. The integration of agricultural modernization and information technology is helpful for improving the level of intelligent agricultural production and network management [38,39]. Industrialization provides an economic foundation and development power for urbanization, whereas urbanization provides element agglomeration and a broad demand market. The interactive development of industrialization and urbanization can improve the efficiency of agricultural production, thus promoting agricultural modernization which in turn fosters the agglomeration of the agricultural population in cities and towns. Therefore, the deep integration of informatization and industrialization, positive interaction between industrialization and urbanization, and coordination between urbanization and agricultural modernization are the inherent requirements and basic laws of the modernization process in today's world.

Xi Jinping proposed Chinese modernization at the 20th CPC National Congress held in October 2022. One of the important goals is to basically achieve new urbanization, industrialization, informatization, and agricultural modernization by 2035 [40]. The NFM

synchronous development strategy meets the requirements of high-quality and sustainable economic development and is the key to solving the social contradictions facing China, such as irrational industrial structure, unbalanced development, and environmental pollution [41]. Although the existing literature on NFM is rich, it mainly focuses on the following aspects: the study of the development model, its characteristics, and level of new urbanization, industrialization, informatization, and agricultural modernization [42–48]; the study of their interactive relationship from the aspects of industrial informatization, urbanization and industrialization, agricultural informatization, and urban–rural integration [49–54]; with the concept of ecological civilization and sustainable development deeply rooted among people, in recent years many scholars have studied the NFM from the perspectives of clean energy, carbon emissions, and environmental protection [55–62]. However, the following questions arise: What is the theoretical basis of China's New Four Modernizations synchronous development strategy? What is the development level of new urbanization, industrialization, informationization, and agricultural modernization in various regions of China? What is the dynamic pattern of NFM synchronous development in China? Is there any significant regional difference in the synchronous development of NFM in China? If so, will this regional difference converge or diverge over time? As the largest developing country in the world, it is of great theoretical and practical significance to answer the above series of questions regarding China's NFM synchronous development strategy. Unfortunately, owing to the lack of existing literature on China's NFM synchronization strategy, it is not possible to answer the above questions systematically.

Therefore, we selected the data of all 31 provinces in mainland China from 2010 to 2020 and used the entropy method and a convergence model to study the dynamic pattern and evolution trend of China's NFM synchronous development. The remainder of this paper is organized as follows: Section 2 expounds the theoretical basis and interaction mechanism of NFM synchronous development. Section 3 constructs the index system and evaluation model of NFM synchronous development and analyzes the dynamic pattern of China's NFM. Section 4 uses the convergence model to analyze the evolution trend of NFM synchronous development in China. Section 5 presents the conclusions and policy recommendations.

## 2. Theoretical Analysis of NFM Synchronous Development

### 2.1. Theoretical Basis

#### 2.1.1. System Theory

The basic idea of system theory is to regard the research object as a system, and to study the law of the relationship and change in various elements of the system's environment. System theory holds that any system has the basic characteristics of integrity and structure, hierarchy and relevance, dynamic balance, and timing [63]. First, the core idea of system theory is integrity; that is, any system is a unified whole according to a certain level and structure. The nature of the system is not possessed by each isolated part, and its function is greater than the sum of its parts [64]. Second, the structural features of a system mean that any system, regardless of its size, has its own structure. A reasonable structure can push the system to develop in a positive and orderly direction, whereas an unreasonable structure will cause the system to develop in a negative and disorderly direction. Third, any system has a hierarchy: from low level to high level, concrete to abstract, simple to complex, and so on. Finally, the dynamic balance of a system means that any system is constantly developing through the processes of emergence, development, and extinction; its balance is relative and phased. The concept of NFM synchronous development is an important embodiment of system theory in practice. As the subsystem of NFM, new urbanization, industrialization, informatization, and agricultural modernization are interconnected, interacted, and influenced by each other; they are an organic whole. The concept of NFM synchronous development also reflects the relevance and hierarchy of internal elements in social development systems, emphasizing the coordination of various subsystems' development.

### 2.1.2. Circular Cumulative Causation Theory

Within the perspective of system theory, famous economists such as Murdal and Kaldo proposed the circular cumulative causation theory [65]. They believe that there is a cyclic cumulative effect among the subsystems in a dynamic economic system. When an initial change occurs in a subsystem, it causes changes in other subsystems. This in turn strengthens the changes in the original system, causing the economic system to continue to develop in the direction of the initial change, thus forming a trend of causal cycle accumulation. The NFM examined in this study follows the circular cumulative causality theory. When a subsystem stimulates its development owing to some factors, it will break the original structure and balance, causing changes in the other "three modernizations", which will successively strengthen the development of the subsystem. Specifically, an improvement at the subsystem development level will affect the development of other subsystems, but this effect has a lag. The original balance is broken, and the degree of system integration (measuring the system's synchronization level) is reduced. However, with the development of other subsystems, the degree of system integration gradually increases and reaches a new higher balance. In other words, in the rising stage of system development, its degree of integration is constantly "falling-rising-falling", but its development level is rising.

### 2.1.3. Theory of the Relationship between the Government and the Market

According to the theory of the relationship between the government and the market, government and market are the "two hands" of the modern market economic system [66]. The government is the "visible hand" of the market economy. It controls the allocation of resources mainly through administrative measures that have a significant impact on the market economy. The market is an "invisible hand" that allocates resources mainly through supply and demand, price, and competition. Evidently, the government and market have different mechanisms and measures of resource allocation. The market regulates production through the price mechanism and the survival of the fittest through the competition mechanism to achieve reasonable and orderly production and consumption and maximize the use of various resources. The market determines resource allocation, which is the basic value law of the market economy. NFM is an important system of the social economy, and the government and the market have a considerable impact on it. The market mechanism is only efficient when information is complete. However, real-world information is incomplete, and the market mechanism will fail. The development of informatization can effectively promote the flow and exchange of information and significantly alleviate the failure of the market mechanism.

In addition, the effects of circular cumulative causality theory and government market relations theory on NFM are interrelated rather than isolated. Specifically, under the effect of market forces, one subsystem of NFM cumulative growth will have diffusion and reflux effects on the other systems. On the one hand, the diffusion effect will have a positive effect on the development of other systems. On the other hand, the reflux effect will make capital, technology, and other favorable conditions of other subsystems return to this subsystem. Diffusion and return effects are often unequal, depending on the development speed of the system under the influence of market forces. The result is that the weak are weaker and the strong are stronger, which causes the gap between the systems to become increasingly larger. Therefore, if we want to achieve the synchronous development of new urbanization, industrialization, informatization, and agricultural modernization, we cannot simply rely on the market mechanism but also on the supervision and regulation of the government.

### 2.2. Analysis of the NFM Interaction Mechanism

First, industrialization is the driving force of urbanization development, and urbanization is the carrier of industrialization development [67]. In the process of industrialization, industry has accelerated, and the industrial structure has been continuously optimized and upgraded. The proportion of secondary and tertiary industries will gradually increase,

and the surplus rural labor force will continue to transfer to cities and towns, thus providing an impetus for urbanization development [68]. Further, urbanization as a carrier of industrialization provides a large number of human resources for industrialization through the aggregation effect; the huge consumer market formed as a result stimulates the development of industrialization [69]. The formation of the industrial chain requires the diversified services provided by urbanization to reduce investment costs and improve economic efficiency.

Second, industrialization and informatization are deeply integrated. Industrialization is the basis of informatization, which consecutively accelerates the industrialization process [70]. Informatization occurs when industrialization reaches a certain stage. It can be said that there is no informatization without industrialization. The information infrastructure and hardware on which informatization depends need the support of industrialization. On the other hand, informatization can optimize the allocation of production factors and improve industrial efficiency by enhancing the transparency of market information and reducing the impact of information asymmetry [71].

Finally, the development of new urbanization, industrialization, and information technology can promote the improvement of agricultural modernization which provides production factors and material guarantees for the other three modernizations [72]. Agricultural modernization requires traditional agriculture to achieve industrial scale and intensification using advanced agricultural production and management concepts, agricultural science, and information technology under the development of new urbanization, industrialization, and information technology. Furthermore, agricultural modernization provides a material basis for industry, such as providing the necessary raw materials and primary products for the industrial sector [73]. With the application of mechanization in agriculture, the productive efficiency of the agricultural sector will be greatly improved. This will inevitably release a large amount of surplus labor, thus providing conditions for the expansion of new urbanization, industrialization, and informatization.

In summary, new urbanization, industrialization, informatization, and agricultural modernization are organic entities that are mutually interconnected, interactive, and promoted [74]. Among them, urbanization is the carrier, industrialization is the engine, informatization is the accelerator, and agricultural modernization is the basis of the other "three modernizations". The NFM synchronous development strategy utilizes their respective advantages through the market, achieving mutual penetration, mutual promotion, and overall consideration. Eventually, a development pattern will be formed in which cities lead villages, industry and agriculture benefit each other, and urban and rural areas are integrated to achieve sustainable socio-economic development (See Figure 1).

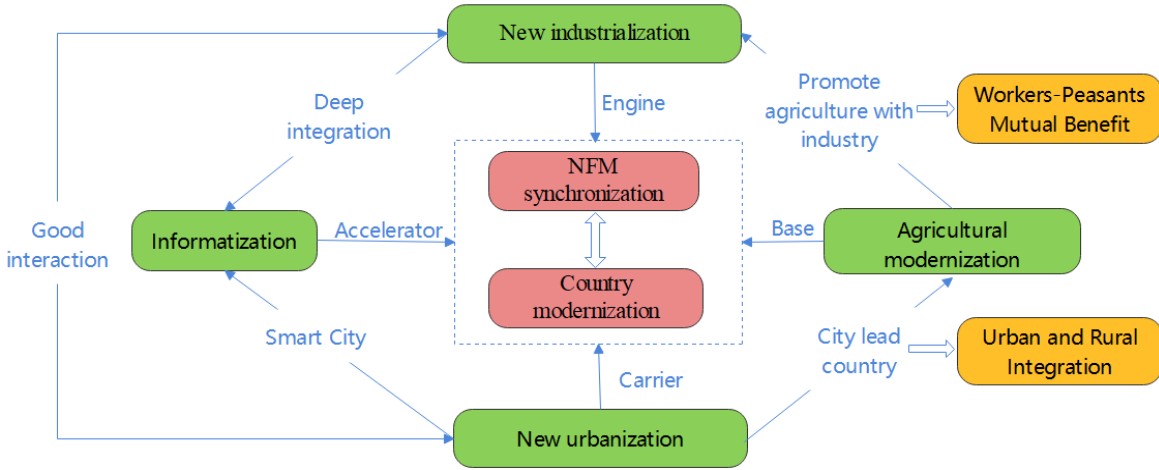

**Figure 1.** Conceptual framework of the NFM interaction mechanism.

## 3. The Measurement and Dynamic Pattern of the China's NFM Synchronous Development Level

*3.1. Indicator System and Data Source*

3.1.1. Indicator System and Description

Referring to existing relevant research [26,42,48,54,69,75,76], and according to the connotation and characteristics of new urbanization, industrialization, informatization, and agricultural modernization, we built an evaluation index system for NFM synchronous development from the perspective of sustainability, scientific and technological innovation, and ecological environment. As shown in Table 1, the evaluation index system includes 4 subsystems of the NFM and 22 original indicators. The indicators are described as follows:

**Table 1.** Indicator system for measuring China's NFM synchronous development level.

| Subsystem | Original Indicators |
|---|---|
| Urbanization (UR) | UR1: Urbanization rate (%) <br> UR2: Per capita disposable income of urban residents (RMB/person) <br> UR3: Urban unemployment rate (%) <br> UR4: Per capita park green area (square meters) <br> UR5: Number of public transportation vehicles (per 10,000 people) <br> UR6: Urban health technicians (per 1000 people) |
| Industrialization (IND) | IND1: Industrialization rate (%) <br> IND2: Industrial employment rate (%) <br> IND3: Industrial labor productivity (%) <br> IND4: Per capita industrial R&D expenditure (RMB) <br> IND5: Utilization rate of industrial solid waste (%) |
| Informatization (INF) | INF1: Internet penetration (%) <br> INF2: Mobile phone penetration (%) <br> INF3: Per capita mobile switch capacity (ten thousand households) <br> INF4: Software industry income (ten thousand yuan) <br> INF5: Number of domain names <br> INF6: Number of webpages |
| Agricultural modernization (AG) | AG1: Disposable income of rural residents (per capita) <br> AG2: Electricity consumption of rural residents (per capita) <br> AG3: Rural mechanical power (per capita) <br> AG4: Rural effective irrigation area (per capita) <br> AG5: Number of rural health technicians (per 1000 people) |

(1) New urbanization. The core of new urbanization is people-oriented. It not only refers to the growth of the urban population and the expansion of urban areas, but also includes the improvement of urban environmental quality and residents' quality of life. Therefore, this study mainly selects urbanization indicators from population, economy, ecological environment, urban facilities, social life, etc. The urbanization rate is the most commonly used indicator to measure the urbanization level, reflecting the dynamic process of population agglomeration in cities and towns, i.e., population urbanization [42]. The per capita disposable income of urban residents and the urban unemployment rate reflect the economic employment and living quality of urban residents, i.e., economic urbanization [69]. The per capita park green area and the number of public transportation vehicles reflect the urban greening level and environmental protection ability. They measure ecological urbanization and infrastructure urbanization. Urban health technicians reflect the medical levels of urban people and measure social urbanization;

(2) New industrialization. New industrialization is a road of industrialization with high scientific and technological levels, good economic benefits, low resource consumption, less environmental pollution, and full utilization of the advantages of human resources. Therefore, selecting new industrialization indicators should not only consider the industrialization development level, but also industrial efficiency, scientific and technological innovation, resources, and environment from the perspective of sustainability. The in-

dustrialization rate is the proportion of industrial added value to GDP [75], reflecting the proportion of industry in the national economy. The industrial employment rate is the proportion of employment in secondary industry to total employment. It reflects the basic characteristics of industrialization. Industrial labor productivity refers to the industrial value added created by the labor force during a certain period, which reflects the economic benefits of the labor input in the industrial sector [54]. Per capita industrial R&D expenditure reflects the industrial R&D investment of enterprises in a certain region and measures the sustainability and innovation of industrial development. New industrialization requires low resource consumption and less environmental pollution. Therefore, this study used the utilization rate of industrial solid waste to measure industrial pollution prevention and environmental protection;

(3) New informatization. The development of information technology can improve information transparency and reduce information blockage, thus improving the efficiency of various production factors. We mainly select information indicators from the information technology application, information infrastructure, and information industry levels. Internet penetration is the proportion of broadband internet access to the total number of people, while mobile phone penetration is the proportion of the total number of mobile phone households to the total number of people. They measure the application level of information technology in China. Per capita mobile switch capacity measures the level of China's information infrastructure. The software industry income measures the industrial informatization level, and the number of domain names and websites measures the social informatization level; together they reflect China's information industry development;

(4) Agricultural modernization. The selection of indicators for agricultural modernization mainly considers agricultural mechanization, agricultural production, and farmers' quality of life. Rural residents' disposable income measures farmers' quality of life, while the electricity consumption of rural residents (per capita) and rural mechanical power (per capita) measure agricultural machinery modernization [26]. Rural effective irrigation areas (per capita) measure agricultural production modernization, and the number of rural health technicians (per 1000 people) measures rural medical care modernization [48].

### 3.1.2. Data Sources and Study Area

The original data for the indicators in this study are from the *China Statistical Yearbook (2011–2021)*, the *Statistical Yearbook of Chinese Provinces* from 2011 to 2021, and the China Internet Network Information Center (http://www.cnnic.net.cn/6/132/index.html, accessed on 8 January 2023). The vast majority of the original indicator data in this paper are complete, with only the information technology indicator INF6 missing in 2010, which belongs to the time series data of a simple missing pattern. A large number of literature [77–79] has proved that linear interpolation has high accuracy in filling such missing data in theory and experiments. Therefore, as much of existing research [80,81], we also use linear interpolation to fill in the missing data. Due to institutional differences and data availability, the study area is all 31 provinces in mainland China (including municipalities directly under the Central Government, excluding Hong Kong, Macao, and Taiwan). According to the interpretation of the National Development and Reform Commission, the division of China's eastern, central, and western regions is based on policy rather than administrative or geographical factors. The divisions in eastern, central, and western China are listed in Table A1.

### 3.2. Evaluation Model and Method

This study employed the entropy method to determine the index weight. The entropy method is objective and avoids the influence of subjective factors; therefore, it can effectively establish the urbanization development index UR, new industrialization development index IND, informatization development index INF, and agricultural modernization development index AG. Finally, we built a degree of fusion model based on the minimum variation coefficient to measure the synchronous development level of NFM in China.

### 3.2.1. Development Index of the NFM Subsystem

The entropy method, which is derived from thermodynamics, is an objective method for determining the weight according to the amount of information contained in the data [82]. It is often used for multi-index evaluation. Specifically, the entropy method determines the weight according to the degree of variation in the indicator data. The greater the degree of difference between the indicators, the more information it provides, the more important it is in the comprehensive evaluation, and the greater its weight. In contrast, its weight is smaller. Thus, the entropy method is the most commonly used measurement model to determine the weight of indicators according to data characteristics. Assume that there are $m$ regions and $n$ indicators to form the original data matrix $X = \left(x_{ij}\right)_{mn}$:

$$X = \begin{bmatrix} x_{11} & x_{12} & \cdots & x_{1n} \\ x_{21} & x_{22} & \cdots & x_{2n} \\ \cdots & \cdots & \cdots & \cdots \\ x_{m1} & x_{m2} & \cdots & x_{mn} \end{bmatrix} \tag{1}$$

(1) To eliminate the impact of different dimensional and attribute data on the comprehensive evaluation, it is necessary to standardize the original data. For the positive indicators, we have

$$z_{ij} = \frac{x_{ij} - \min\left(x_j\right)}{\max\left(x_j\right) - \min\left(x_j\right)} \tag{2}$$

For negative indicators, there are

$$z_{ij} = \frac{\max\left(x_j\right) - x_{ij}}{\max\left(x_j\right) - \min\left(x_j\right)} \tag{3}$$

where $z_{ij}$ is the standardized value of the $i$th evaluation object on the $j$th indicator, $x_{ij}$ is the original value, and $\max(x_j)$, $\min(x_j)$ are the maximum and minimum values of the $j$th indicator data, respectively.

(2) The proportion of the $i$th evaluation object on the $j$th indicator is calculated as

$$P_{ij} = \frac{x_{ij}}{\sum_{i=1}^{m} x_{ij}} \tag{4}$$

(3) The expression of entropy $e_j$ of the $j$th indicator is

$$e_j = -k \sum_{i=1}^{m} \left(P_{ij} . \ln P_{ij}\right) \tag{5}$$

where $k = 1/\ln(m)$, and $m$ is the number of evaluation object.

(4) The expression of the $j$th index weight is

$$W_j = \left(1 - e_j\right) / \sum_{j=1}^{n} \left(1 - e_j\right) \qquad W_j \in (0, 1) \tag{6}$$

(5) Finally, the scores for each evaluation object can be obtained and indexed.

$$F_i = 100 \sum_{j=1}^{n} \left(z_{ij} . W_j\right) \qquad F_i \in [0, 100] \tag{7}$$

Using the entropy method, we obtained the weights of China's NFM subsystem indicators, as shown in Table A2. We can then calculate the new urbanization index UR, industrialization index IND, informatization index INF, and agricultural modernization index AG for all regions in China from 2010 to 2020.

### 3.2.2. The NFM Comprehensive Development Index

Under the concept of NFM synchronous development, the new urbanization development index UR, industrialization index IND, agricultural modernization index INF, and informatization index AG are equally important. Therefore, we define the NFM comprehensive development index *T* as the mean value of subsystem index:

$$T = \tfrac{1}{4}[\text{UR} + \text{IND} + \text{INF} + \text{AG}] \tag{8}$$

### 3.2.3. Degree of Fusion Model Based on Minimum Variation Coefficient

The essence of China's NFM synchronous development concept is "synchronization" where the core questions are "What is synchronization?" and "How to synchronize?". This study considers that system synchronization means that each system has coordination or fusion, so we introduce the concept of degree of fusion. We used the variation coefficient in mathematical statistics to define degree of fusion. The variation coefficient is defined as the ratio of the array standard deviation to its corresponding mean value, which can accurately measure the dispersion of random variables [83,84]. The smaller the variation coefficient, the more concentrated (fused) the group of data, and the more discrete it is.

Let the two subsystems be denoted as *f(x)* and *f(y)*, respectively, with positive values. Their mean and standard deviation are $\mu$ and $\delta$, respectively. The variation coefficient $C_v$ is defined as

$$C_v = \frac{\delta}{\mu} = \frac{\delta}{\frac{f(x)+f(y)}{2}} = \sqrt{2\left(1 - \frac{f(x)f(y)}{\left[\frac{f(x)+f(y)}{2}\right]^2}\right)} \tag{9}$$

The smaller the variation coefficient $C_v$, the better the fusion of the system. For the above formula, the necessary and sufficient conditions for the minimum value of $C_v$ are

$$C = \frac{f(x)f(y)}{\left[\frac{f(x)+f(y)}{2}\right]^2} \tag{10}$$

where *C* is the maximum value. By extending the two subsystems to *n* subsystems $f_1(x), f_2(x) \ldots f_n(x)$, we can obtain the degree of fusion model based on the minimization of the variation coefficient, namely

$$C = \frac{f_1(x)f_2(x)\cdots f_n(x)}{\left[\frac{f_1(x)+f_2(x)+\cdots f_n(x)}{n}\right]^n} \quad C \in [0,1] \tag{11}$$

where $f_1(x), f_2(x) \ldots f_n(x)$ are the comprehensive evaluation functions of the subsystems, *n* is the adjustment coefficient, and *C* is the degree of fusion. *C* is a quantitative index used to measure the quality of synchronization between systems: the greater the value, the more synchronization between systems.

Considering NFM synchronous development, we take *n* = 4, then the NFM's degree of fusion model is

$$C = \frac{\text{UR} \times \text{IND} \times \text{INF} \times \text{AG}}{\left[\frac{\text{UR}+\text{IND}+\text{INF}+\text{AG}}{4}\right]^4} \quad C \in [0,1] \tag{12}$$

where UR, IND, INF, and AG are the new urbanization development, industrialization, agricultural modernization, and informatization indices, respectively. *C* is the degree of fusion between new urbanization, industrialization, informatization, and agricultural modernization.

### 3.2.4. The NFM Synchronous Development Degree Model

In practice, if the development level of each subsystem of a research object is extremely backward, the degree of fusion between them will also be extremely high. Therefore, using the degree of fusion alone cannot depict the information of the system synchronization and development simultaneously. To compensate for the lack of degree of fusion, we also consider the comprehensive development index *T* and the degree of fusion *C*. Therefore,

we propose the concept of the degree of NFM synchronous development, which comprehensively considers the development and synchronization levels. The calculation formula is as follows:

$$D = \sqrt{100 \times C \times T} \quad\quad D \in [0, 100] \quad\quad\quad (13)$$

where $D$ is the degree of synchronous development of the research object which is a quantitative indicator for measuring the NFM's synchronous development level; $C$ is the NFM's degree of fusion; and $T$ is the NFM's comprehensive development index. According to the value range, this paper uses the idea of equality to divide the development stage of NFM, as shown in Table 2.

**Table 2.** Division of development stages.

| Development Stage | Initial | Middle | Middle-late | Late |
|---|---|---|---|---|
| Range | [0, 25] | (25, 50] | (50, 75] | (75, 100] |

### *3.3. Results Analysis*

We used the open source statistical programming language R to calculate the subsystem development index (UR, IND, INF, AG), comprehensive development index $T$, degree of fusion $C$, and synchronous development level $D$ of 31 provinces in China (excluding Hong Kong, Macao, and Taiwan) from 2010 to 2020. We then analyzed the dynamic patterns of China's NFM according to the measurement results.

### 3.3.1. Dynamic Pattern of the NFM Subsystem in China

Figure 2 shows the development level of China's NFM subsystem from 2010 to 2020. First, the development of China's new urbanization, industrialization, informationization, and agricultural modernization was on the rise. Second, the development level of informatization and agricultural modernization significantly lagged behind new urbanization and industrialization, indicating that the imbalance and non-synchronization of China's NFM internal development was very prominent. Finally, China's new industrialization had the largest fluctuations, followed by informatization; new urbanization and agricultural modernization had the smallest fluctuations. This may be because new industrialization and informatization are sensitive and easily affected by the economic cycle, technological innovation, and other factors, whereas new urbanization and agricultural modernization are relatively stable.

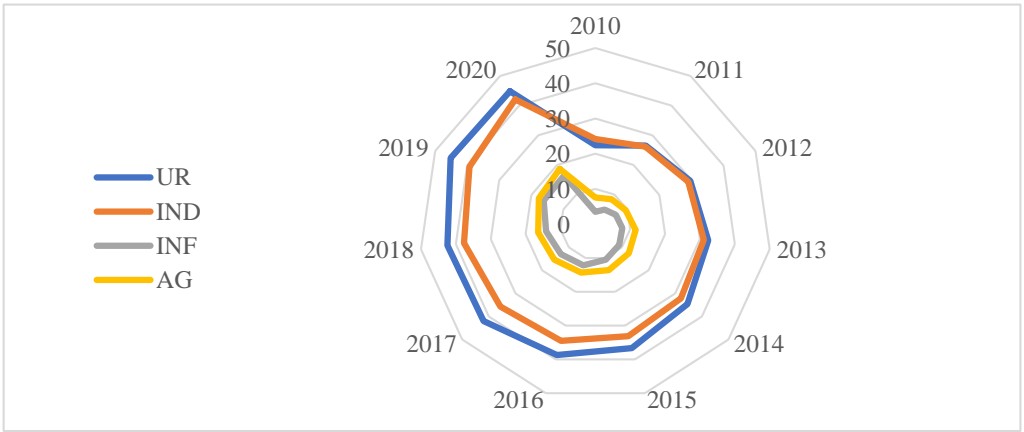

**Figure 2.** Development level of China's new urbanization, industrialization, informatization, and agricultural modernization from 2010 to 2020.

As shown in Figure 3a, China's new urbanization development index grew from 22.43 in 2010 to 44.86 in 2020, with an average annual growth rate of 7.12%; this is considered a rapid development rate. It is currently in the middle stage of urbanization. According to China's National Bureau of Statistics, China's urbanization rate reached 64.72% in 2021, 8.72 percentage points higher than the world average with significant progress in population, land, and facility urbanization. From a regional perspective, the development level of new urbanization in the central and western regions lagged significantly behind that of the eastern regions and was lower than the national average. The new urbanization development index in the eastern region reached 51.91 in 2020, entering the middle-late stage of urbanization. Since the reform and its opening up, China has adopted a policy of prioritizing the development of the Yangtze River Delta, the Pearl River Delta, and other eastern coastal regions. With a developed economy, convenient transportation, and higher population density, the eastern region has a better foundation for urbanization development and attracts a large number of talents, capital, and technologies. At the same time, the development of China's new urbanization still faces many problems. For example, many peasants have difficulty integrating into cities, thus causing a series of problems in education, medical care, and society. In the future, we should focus on the development of ecological urbanization, social urbanization, and living urbanization; improving urban infrastructure, medical care, and public services; creating a green, comfortable, healthy and livable urban environment; and realizing the high quality development of new urbanization.

As shown in Figure 3b, China's new industrialization index grew from 24.15 in 2010 to 42.11 in 2020, with an average annual growth rate of 5.72% and is currently in the middle stages of industrialization development. From a regional perspective, the development of new industrialization has shown a decreasing spatial pattern in the eastern, central, and western regions. The new industrialization index of the eastern region reached 60.96 in 2020, indicating that the eastern region entered the middle-late stages of industrialization. Relying on the advantages of coastal transportation, the eastern region has a high degree of openness, developed international trade, and a leading position in industrialization. However, the development level of new industrialization within the eastern region is quite different, and the polarization phenomenon is relatively serious. In 2020, Shanghai, with the highest index of new industrialization (91.33), was far higher than Hainan province, with the lowest index (21.15). In 2020, the new industrialization indices in the central and western regions were 36.19 and 28.78, respectively; this was during the middle stage of industrialization. The central region has a better industrialization foundation, but the new industrialization development level has lagged behind that in the eastern region due to relatively backward ideologies, imperfect market mechanisms, and serious brain drain. On the other hand, with the increasing costs of enterprises in the eastern region, a large number of manufacturing industries have shifted to the central region, such as Foxconn and other large enterprises landing at Zhengzhou Airport. In addition, the central region has a high population density, large market potential, and convenient transportation location. The development potential of new industrialization in the central region is still significant. The western region is vast and sparsely populated; it has inconvenient transportation which makes transportation costs higher and the market potential insufficient. Therefore, the development of new industrialization in the western region was the most backward.

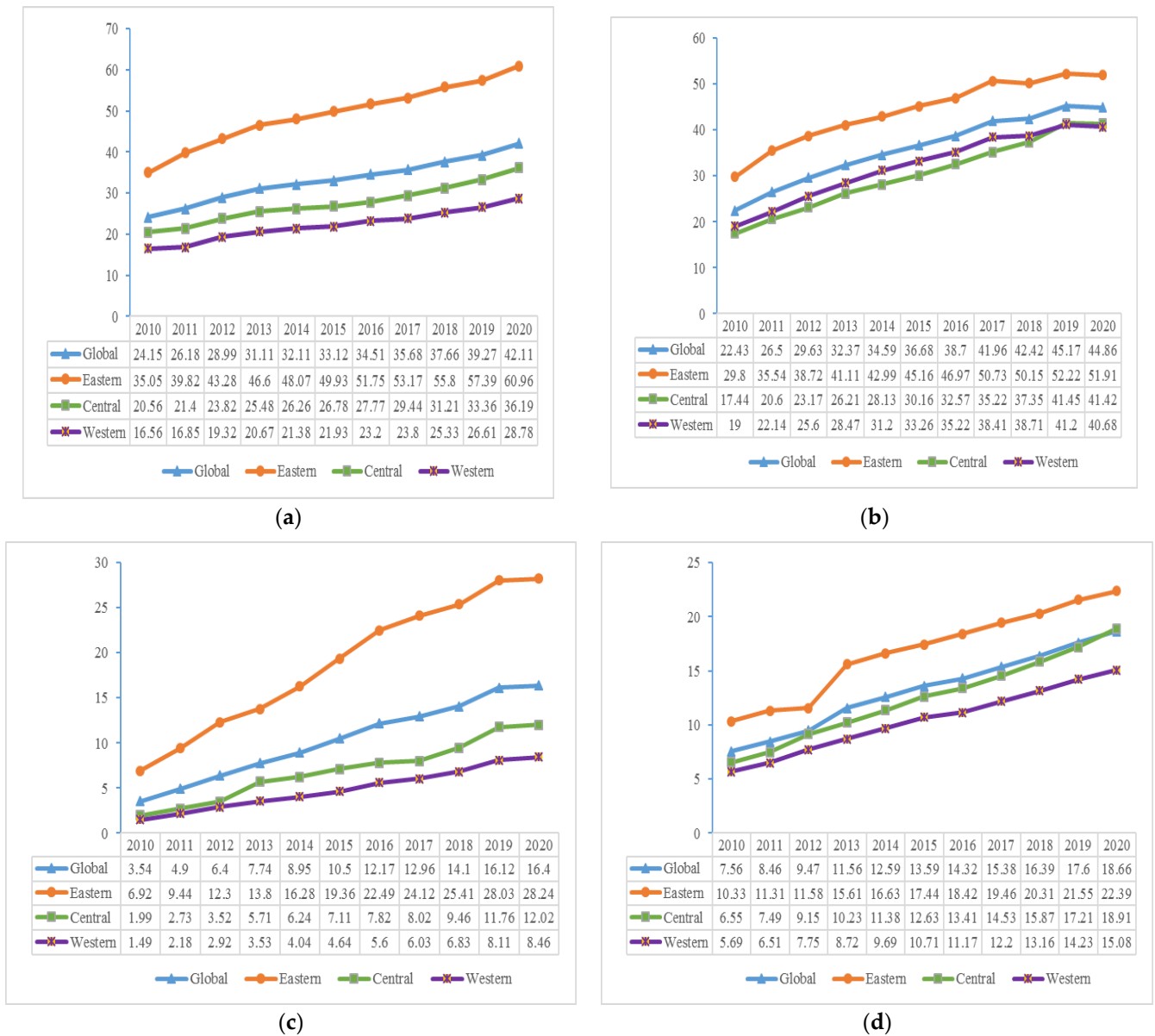

**Figure 3.** Development level of the NFM subsystem in the east, central, and west of China from 2010 to 2020: (**a**) New urbanization index; (**b**) New industrialization index; (**c**) Informatization index; (**d**) Agricultural modernization index.

As Figure 3c shows, China's informatization index increased from 3.54 in 2010 to 16.4 in 2020, with an average annual growth rate of 16.57%. This indicates that China's informatization foundation was weak and the overall development level was low, but the growth rate was high; it was in the initial stage of rapid development. From a regional perspective, the informationization index of the eastern region in 2020 was 28.24, i.e., in the middle stage of information development, whereas the central and western regions were still in the initial stage of information development. The gap between informationization development in the eastern region and the central and western regions is obvious. This is because the eastern region has a perfect market, abundant capital, technology, talent, and a relatively open entrepreneurial environment. At present, information technology service is a direction for industrial upgrading in the future. With the help of "internet +", we should vigorously promote the depth of informatization and industrialization, make the exchange of information faster, and improve the efficiency of industrial and market development.

China's informatization is undergoing explosive development, and informatization will be an important force to promote NFM synchronous development in the future.

As shown in Figure 3d, China's agricultural modernization index increased from 7.56 in 2010 to 18.66 in 2020, with an average annual growth rate of 9.46%; this was the initial stage of rapid development. From a regional perspective, the agricultural modernization indices of the eastern, central, and western regions in 2020 were 22.39, 18.91, and 15.08, respectively; they were all in the initial stages of agricultural modernization development. The level of agricultural modernization in the eastern, central, and western regions continued to decrease, but the gap between the three regions was not large. These measurement results reflect the current situation of China's agricultural development. Although grain output ranks first in the world, the agricultural production level is backward, scientific and technological content is not high, and extensive management is still very common. China's agricultural production is still a basic household unit, and it is difficult to form large-scale operations. Planting and breeding mainly rely on traditional experiences and lack scientific and technical guidance. The construction of farmland infrastructure is stagnant, and productive services are insufficient for livestock distribution. The overall cultural quality of farmers is not high, and the loss of young and middle-aged farmers is serious. Finally, agricultural modernization has become a weak link in the development of NFM, which needs to be upgraded and urgently improved.

Figure 4 shows the dynamic evolution pattern of new urbanization, industrialization, informationization, and agricultural modernization in Chinese provinces in recent years. Based on the 2010, 2015, and 2020 values of indicators UR, IND, INF, and AG, we have plotted a thermal map with larger values represented by a darker shade of red. First, in terms of the new urbanization development stage, most regions in China in 2010 were in the initial stage (up to 22 provinces); eight provinces were in the middle stage; only Beijing (52.52) was in the middle-late stage; and no region reached the late stage. However, by 2020, the vast majority of China's regions were in the middle stage of new urbanization (24 provinces), and the number of provinces in the middle-later stages reached six with Beijing (77.92) reaching the later stage. Second, in the past decade, in terms of the new industrialization development stage, the number of provinces in the initial stage decreased from 21 to 8, and the number in the middle stage increased from 8 to 14. The number in the middle-late stage increased from 2 to 6, and the number in the late stage increased from 0 to 3. Third, all 31 provinces in China in 2010 were in the initial stage of new informatization development. By 2020, the number of provinces in the initial stage of new informatization was 26, while the number of provinces in the middle, middle-late, and late stage was 3, 1, and 1, respectively. Finally, in terms of agricultural modernization, China's 31 provinces were in the initial stage in 2010. By 2020, 27 provinces were in the initial stage, 3 provinces were in the middle stage, and 1 province was in the middle-late stage.

In short, in terms of temporal trends, new urbanization, industrialization, informatization, and agricultural modernization are evolving to higher developmental stages in China's 31 provinces. In terms of spatial patterns, the capital and the southeast coastal provinces have higher development levels of new urbanization, industrialization, and informatization; the provinces with higher development levels of agricultural modernization are mainly distributed in the northeast and southern regions of China's main grain production areas.

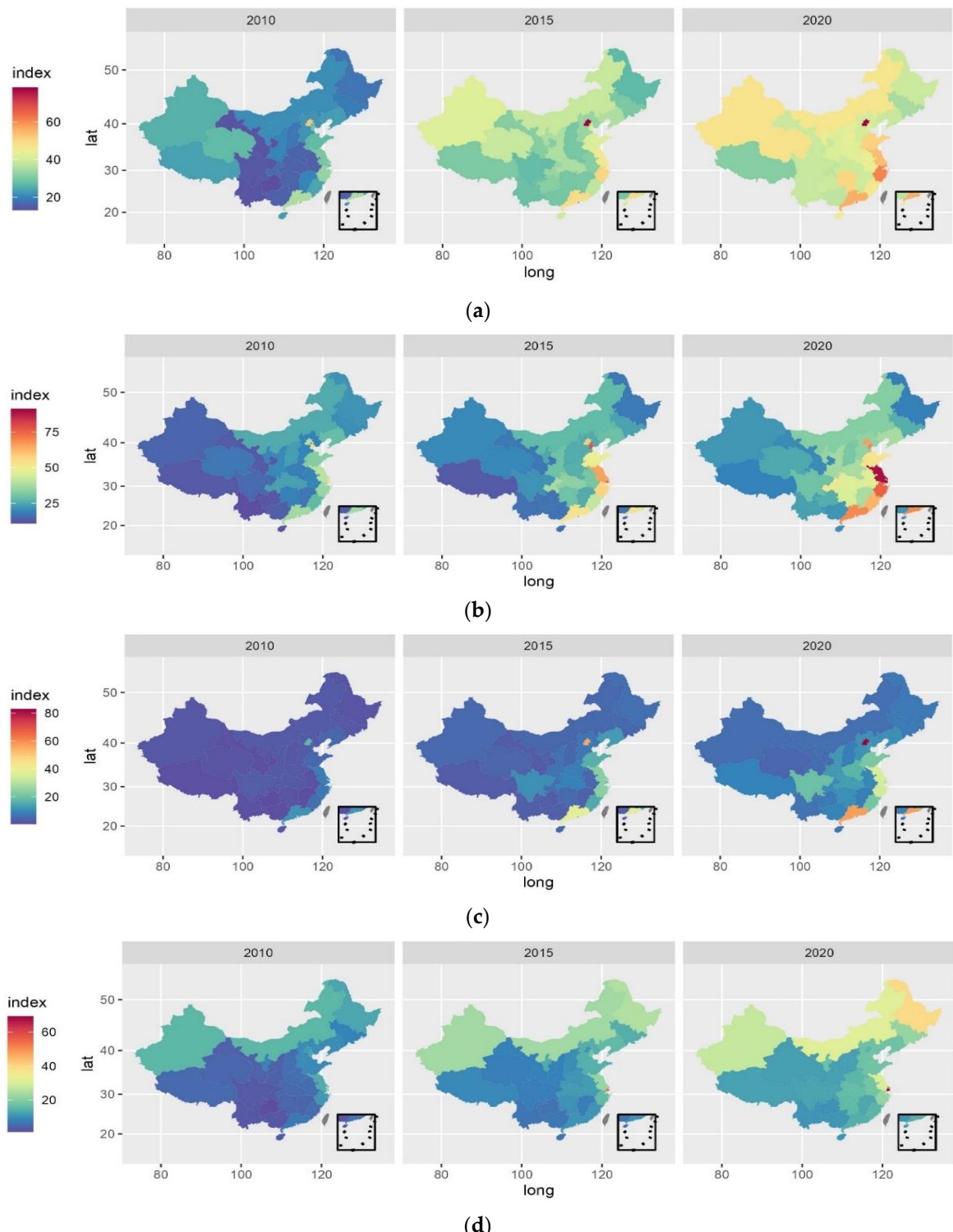

**Figure 4.** Dynamic evolution pattern of China's new urbanization, industrialization, informatization, and agricultural modernization: (**a**) Trend map of China's urbanization evolution; (**b**) Trend map of China's industrialization evolution; (**c**) Trend map of China's informatization evolution; (**d**) Trend map of China's agricultural modernization evolution.

3.3.2. Dynamic Pattern Analysis of China's NFM Synchronous Development

Figure 5 reports the NFM synchronous development level of China and the three regions, i.e., eastern, central, and western. From a global perspective, China's NFM synchronous development index was 18.62 in 2010 and 38.75 in 2020, indicating that the

overall level of China's NFM synchronous development has improved significantly from the initial stage to the middle stage. From a regional point of view, the NFM synchronous development index in the eastern region was 27.29 in 2010, just entering the middle stage. However, the NFM synchronous development index reached 47.67 in 2020, which was about to enter the middle stage. The NFM synchronous development index in the central region increased from 15.13 in 2010 to 36.8 in 2020, jumping from the initial stage to the middle stage. In the past ten years, the NFM synchronous development level of the western region has also jumped from the initial stage to the middle stage. From the perspective of development trends, the synchronous development level of NFM in China and the three regions has steadily increased over time. However, the eastern region was significantly higher than the global, central, and western regions, showing a spatial pattern of east > global > central > western. Therefore, although China's NFM synchronous development level was steadily increasing, the problem of unbalanced and unsynchronized development was prominent.

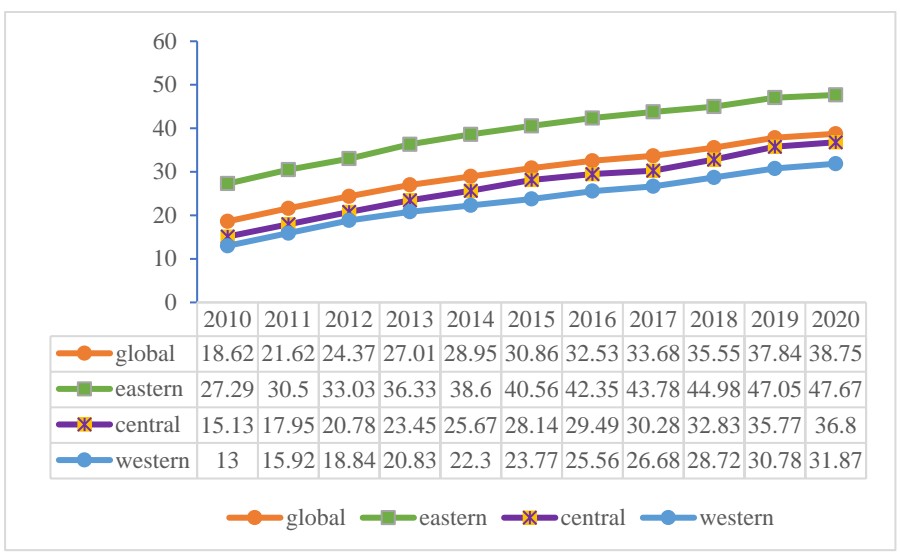

**Figure 5.** The NFM synchronous development index in China and the three regions from 2010 to 2020.

Based on the NFM synchronous development level (namely D value) of the 31 provinces in 2010, 2015, and 2020, we created a heat map (see Figure 6). The darker the red color, the larger the corresponding value. According to Figure 6, the NFM synchronous development level in China's 31 provinces has improved over time, but there are significant spatial differences. Specifically, in 2010, only six provinces in China were in the middle stage of NFM synchronous development, namely Beijing (39.54), Shanghai (35.9), Jiangsu (33.61), Zhejiang (30.35), Guangdong (30.30), and Liaoning (25.36), while the other 25 provinces were all at an initial stage. However, five provinces in China reached the middle-late stage of NFM synchronous development in 2020, namely Shanghai (67.66), Jiangsu (60.33), Zhejiang (55.60), Beijing (53.40), and Guangdong (51.42). Meanwhile, 25 provinces are in the middle stage and only Qinghai (23.68) is still at an initial stage. It is not difficult to see that the provinces with high levels of NFM synchronous development in China are mainly concentrated in the economically developed southeastern coastal areas. This region is characterized by convenient transportation, mature markets, relatively complete social systems, obvious advantages in capital, science and technology, and talent; it is also the most innovative and dynamic region in China.

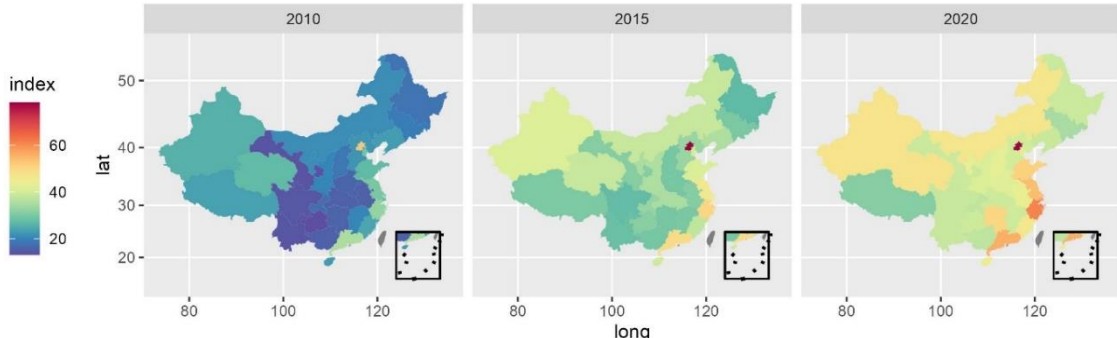

**Figure 6.** Dynamic evolution pattern of NFM synchronous development in China.

## 4. Convergence Analysis of Regional Differences in China's NFM Synchronous Development

### 4.1. Econometric Model

We have analyzed the dynamic pattern of China's NFM synchronous development in detail. The results showed that there were significant differences in NFM synchronous development across China at present, with different levels and stages of development among regions. Therefore, will regional differences in the synchronous development level of China's NFM tend to converge or diverge over time? Will the backward areas narrow the gap with the advanced areas at a faster rate of development, and eventually achieve steady-state convergence of NFM synchronous development in various regions of China? In this study, $\sigma$-convergence and $\beta$-convergence models were introduced to analyze the convergence of China's NFM synchronous development, providing a basis for balanced regional development. This is of great significance for understanding the future evolution trend of China's NFM synchronous development.

#### 4.1.1. $\sigma$-Convergence Model

$\sigma$-Convergence is expressed by the dispersion of the cross section of panel data. Commonly used methods for testing $\sigma$-convergence include standard deviation and coefficient of variation [85]. Considering comparability, the coefficient of variation is more commonly used, and the calculation formula is

$$\sigma_t = \sqrt{\frac{\sum\limits_{i=1}^{n}\left(lny_{it} - \frac{1}{n}\sum_{i=1}^{n}lny_{it}\right)^2}{n}} \bigg/ \frac{1}{n}\sum_{i=1}^{n}lny_{it} \tag{14}$$

where $y_{it}$ is the NFM synchronous development index of the $i$th region in the $t$th year, and $n$ is the number of regions. If it decreases with time, that is, $\sigma_{t+1} < \sigma_t$, then $\sigma$-convergence exists, thus indicating that the gap in the NFM synchronous development level between regions shrinks. If it increases or does not change with time, the $\sigma$-convergence does not exist, indicating that the gap in the NFM synchronous development level between regions does not shrink and may even be expanding.

#### 4.1.2. $\beta$-Convergence Model

According to the definition of $\beta$-convergence by *Barro* and *Sala-i-Martin* [86,87], in the process of China's NFM synchronous development, if the lagging region grows faster than the advanced region, and the former can catch up with the latter after a period of time, we say that there is $\beta$-convergence. $\beta$-convergence examines whether regions with lower initial development levels have higher growth rates than regions with higher development levels. In other words, the growth rate of NFM synchronous development in different regions is negatively correlated with initial development levels. According to the different conditions of convergence, $\beta$-convergence includes absolute $\beta$-convergence and conditional $\beta$-convergence.

The implicit assumption of absolute $\beta$-convergence is that all regions have completely consistent basic characteristics. Thus, NFM synchronous development also has a completely consistent growth path and stable equilibrium. Therefore, in a system with the same basic characteristics, the growth rate of each region is inversely proportional to its distance from its steady state. The test model is as follows:

$$ln\left(\frac{y_{i,t+1}}{y_{it}}\right) = \alpha + \beta ln(y_{it}) + \varepsilon_{it} \tag{15}$$

where $y_{i,t+1}$ and $y_{it}$ represent the NFM synchronous development index of the $i$th region in year $t + 1$ and $t$, respectively; $\alpha$ is a constant term; and $\varepsilon_{it}$ is a random disturbance term. $\beta$ is a convergence coefficient, which is the core coefficient of concern. When $\beta$ is less than 0 and significant, $\beta$-convergence occurs; the larger the absolute value of $\beta$, the faster the convergence rate. When $\beta$ is greater than zero, $\beta$-convergence does not occur. According to the $\beta$-convergence coefficient and time span $T$, the corresponding convergence rate $\theta$ can be calculated as follows:

$$\theta = \frac{-ln(1+\beta)}{T}. \tag{16}$$

$\rho$ is the half-range convergence period, and the calculation formula is

$$\rho = \frac{ln(2)}{\theta}. \tag{17}$$

The absolute $\beta$ convergence mainly considers the effect of the initial level of NFM synchronous development and ignores the effect of other exogenous variables. In fact, there are differences in economic development, education, and technological innovation in each region. Therefore, NFM synchronous development in each region has different steady states. Conditional $\beta$-convergence abandons the assumption that all regions have the same development characteristics and states that exogenous variables have different effects on different regions. In other words, different regions have heterogeneous development characteristics and have different growth paths and steady states. Conditional $\beta$-convergence reflects whether there is a negative correlation between the growth rate and the initial level when the exogenous variables remain unchanged. The test model is as follows:

$$ln\left(\frac{y_{i,t+1}}{y_{it}}\right) = \alpha + \beta ln(y_{it}) + \gamma X_{it} + \varepsilon_{it} \tag{18}$$

where $X$ is an exogenous conditional variable reflecting regional heterogeneity. The relevant conditional variables selected in this study were as follows: (1) GDP per capita is denoted as *gdp* which measures the economic development levels in various regions of China. (2) Per capita education expenditure is denoted as *edu* which is used to measure the education levels in various regions of China. (3) The authorized amount of invention patents. It measures the scientific and technological innovation capacity of various regions in China and is denoted as *tech*.

### 4.2. Analysis of Empirical Results
4.2.1. Empirical Analysis of $\sigma$-Convergence

Figure 7 depicts the $\sigma$ dynamic evolution trend of NFM synchronous development in China and the three regions during the sample period. From a global perspective, the $\sigma$ coefficient shows an obvious and stable declining trend over time, indicating that NFM synchronous development presents a typical $\sigma$-convergence. Therefore, the gap in the level of NFM synchronous development among China's 31 provinces has gradually narrowed over time. From the perspective of evolutionary characteristics, the global $\sigma$ coefficient of China was in an accelerated stage of decline during 2010–2012, and in a stable stage of decline during 2013–2020, with a decrease of 42.98% in the sample period. Second, the $\sigma$ coefficient in the eastern region decreased from 0.28 in 2010 to 0.24 in 2020, showing

a fluctuating downward trend. However, the decline was not obvious, thus supporting the σ-convergence hypothesis to some extent. Third, the σ coefficient in the central region decreased from 0.222 in 2010 to 0.102 in 2020, i.e., a decrease of 54.05%, showing a significant downward trend; this supports the σ-convergence hypothesis. This shows that the gap in the synchronous development level of NFM among the eight provinces in central China gradually narrowed over time. The σ coefficient in the western region decreased the most from 0.375 in 2010 to 0.152 in 2020, reaching 59.47%. The convergence characteristics of the western region are consistent with the global and present a typical σ-convergence; this indicates that the gap between the NFM synchronous development level of the 12 provinces in the western region is significantly narrowing. In short, there was a clear phenomenon of σ-convergence in China, the central and western regions, and to some extent in the eastern region, indicating that the differences in the NFM synchronous development in China and the three regions were on a narrowing trend.

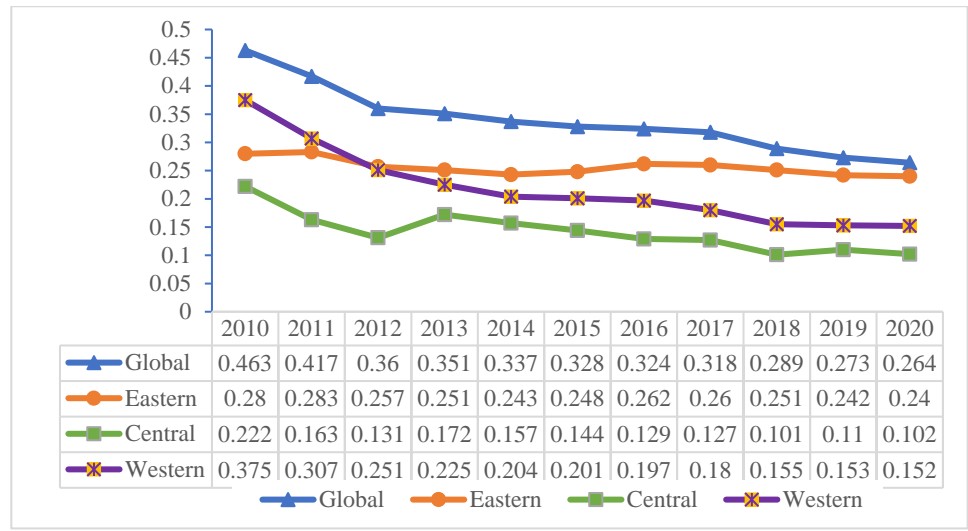

| | 2010 | 2011 | 2012 | 2013 | 2014 | 2015 | 2016 | 2017 | 2018 | 2019 | 2020 |
|---|---|---|---|---|---|---|---|---|---|---|---|
| Global | 0.463 | 0.417 | 0.36 | 0.351 | 0.337 | 0.328 | 0.324 | 0.318 | 0.289 | 0.273 | 0.264 |
| Eastern | 0.28 | 0.283 | 0.257 | 0.251 | 0.243 | 0.248 | 0.262 | 0.26 | 0.251 | 0.242 | 0.24 |
| Central | 0.222 | 0.163 | 0.131 | 0.172 | 0.157 | 0.144 | 0.129 | 0.127 | 0.101 | 0.11 | 0.102 |
| Western | 0.375 | 0.307 | 0.251 | 0.225 | 0.204 | 0.201 | 0.197 | 0.18 | 0.155 | 0.153 | 0.152 |

**Figure 7.** The σ coefficient's evolutionary trend for NFM synchronous development in China and the three regions during 2010–2020.

### 4.2.2. Empirical Analysis of β-Convergence

To examine whether the lagging regions can catch up with the advanced regions at a faster rate and whether the gap between different regions eventually converges over time, we employed the β-convergence model to test NFM synchronous development in China and the three regions.

Panel A in Table 3 reports the test results for absolute β-convergence in the global and the eastern, central, and western regions. It can be seen that the estimated β coefficients are negative at the 1% significance level, indicating that NFM synchronous development, both nationally and in the three regions, is supported by the absolute β convergence mechanism. At the global or regional level, backward provinces are catching up with advanced provinces at a fast speed, and the gap between NFM synchronized development will narrow in the future eventually reaching convergence. From the convergence results, the convergence speed and convergence period of NFM synchronous development are different in different regions. The convergence speed of the central and western regions is significantly faster than that of the global. The convergence period is smaller than that of the global, whereas the convergence speed of eastern regions is significantly lower than that of the global. The convergence period is larger than that of the global. This shows that although the central and western regions lag behind the eastern regions in terms of NFM synchronous development, the development speed is faster, and their development is more balanced. There are two possible reasons for this. First, due to the great differences in geographical location, politics, and economy among the provinces in the eastern region, the unbalanced development of the NFM within the region has become more prominent.

Second, in recent years, to ensure the goal of balanced regional development and common prosperity, the Chinese government has implemented regional strategies such as Western (Region) Development and the Rise of Central China.

**Table 3.** The $\beta$-convergence estimation results of NFM synchronous development in China and the eastern, central, and western regions.

| Variable | Panel A: Absolute $\beta$-Convergence | | | |
|---|---|---|---|---|
| | **Global** | **Eastern** | **Central** | **Western** |
| $\beta$ | −0.1214 *** (0.0086) | −0.0708 *** (0.0151) | −0.1652 *** (0.0196) | −0.1926 *** (0.0165) |
| $\alpha$ | 0.4806 *** (0.0284) | 0.3121 *** (0.0547) | 0.6219 *** (0.0631) | 0.6858 *** (0.0508) |
| Adjusted R$^2$ | 0.3937 | 0.1611 | 0.4710 | 0.5331 |
| Convergence rate (%) | 1.29 | 0.73 | 1.81 | 2.14 |
| Half convergence period | 53.73 | 94.95 | 38.51 | 32.39 |
| N | 310 | 110 | 80 | 120 |
| Variable | Panel B: Conditional $\beta$-convergence | | | |
| | **Global** | **Eastern** | **Central** | **Western** |
| $\beta$ | −0.1878 *** (0.0184) | −0.1576 *** (0.0389) | −0.2076 *** (0.0434) | −0.2283 *** (0.0363) |
| *tech* | 0.0087 ** (0.0042) | 0.0093 (0.0089) | 0.0011 (0.0149) | 0.0086 (0.0059) |
| *gdp* | 0.0627 *** (0.0215) | 0.1080 *** (0.0381) | 0.1236 ** (0.0547) | 0.0181 (0.0389) |
| *edu* | −0.0373 ** (0.0159) | −0.0961 *** (0.0285) | −0.0682 (0.0469) | 0.0097 (0.0281) |
| $\alpha$ | 0.2504 ** (0.1216) | 0.1055 (0.1799) | −0.0395 (0.3252) | 0.4719 (0.2777) |
| Adjusted R$^2$ | 0.4581 | 0.2568 | 0.5135 | 0.5379 |
| Convergence rate (%) | 2.08 | 1.72 | 2.33 | 2.59 |
| Half convergence period | 33.32 | 40.30 | 29.75 | 26.76 |
| N | 310 | 110 | 80 | 120 |

Note: ** $p < 0.05$; *** $p < 0.01$.

Panel B in Table 3 reports the test results of conditional $\beta$-convergence in the global and the eastern, central, and western regions. The estimated coefficient $\beta$ is still negative at the significance level of 1%, that is, there are conditional $\beta$-convergence characteristics for NFM synchronous development nationally and in the three regions. This shows that after controlling the heterogeneous conditions, such as technological innovation, economic development, and education level, China's NFM synchronous development still converges to the steady level. In terms of the convergence speed, as with the absolute $\beta$ convergence, it still shows that east < central < west, but the overall convergence rate is significantly improved. It indicates that the relevant economic and social characteristics accelerate the convergence speed of NFM synchronous development, while shortening the corresponding convergence period. Specifically, the coefficient of the variable *tech* and the

variable *gdp* is significantly positive, indicating that technological innovation and economic development have a close positive relationship with the synchronous development of NFM. This is because technological innovation and economic development have spillover effects which can synergistically promote the development of new urbanization, industrialization, informatization, and agricultural modernization. The coefficient of *edu* is significantly negative, indicating that there is a negative correlation between education level and NFM synchronous development. China's urban–rural dual structure, ideological concepts, and other factors lead to the fact that the improvement in educational levels will prompt a large number of highly qualified talents from rural areas to gather in cities and towns. This will foster the development of industrialization and informatization but not the development of agricultural modernization, thus undermining NFM synchronous development.

## 5. Conclusions and Policy Recommendations

### 5.1. Conclusions

This study expounded the theoretical basis and interaction mechanism of NFM synchronous development in China. We then constructed the index system to evaluate the NFM synchronous development level and used the entropy method to measure the new urbanization, industrialization, information, and agricultural modernization indices of 31 provinces and three regions in China from 2010 to 2020. Moreover, we constructed a degree of fusion model to analyze the dynamic patterns of China's NFM synchronous development. Finally, we conducted an empirical analysis of the convergence of NFM synchronous development in China by using $\sigma$-convergence and a $\beta$-convergence models, which are helpful for correctly understanding the future evolutionary trend of NFM synchronous development in emerging countries such as China. The following conclusions can be drawn based on the empirical analysis results:

(1) In terms of the current situation of NFM development, China's new urbanization and industrialization were in the middle stage, while informationization and agricultural modernization were in the initial stage of rapid development. From a regional perspective, new urbanization and industrialization in the eastern region were in the middle-late stages, informatization was in the middle stage, and agricultural modernization was in the early stage. New urbanization and industrialization in the central and western regions were in the middle stage, whereas informationization and agricultural modernization were in the initial stage;

(2) The level of China's NFM synchronous development continued to improve, but the problem of unbalanced, uncoordinated, and unsynchronized development remained prominent. On the one hand, the development levels of China's new urbanization, industrialization, informationization, and agricultural modernization were quite different, displaying a decreasing pattern. Moreover, the development of informatization and agricultural modernization lagged behind new urbanization and industrialization. This demonstrated that there were obvious problems of unbalanced and unsynchronized development in China's NFM subsystem. In terms of spatial distribution, the NFM synchronous development level in the eastern, central, and western regions of China showed a decreasing spatial pattern in which the eastern region was significantly higher than the central and western regions, thus indicating that the problems of unbalanced and uncoordinated development among regions were prominent;

(3) There was significant $\sigma$-convergence and $\beta$-convergence in China's NFM synchronous development which showed that although the NFM synchronous development level in China was unbalanced and uncoordinated, the backward regions were catching up, and the regional differences may gradually narrow in the future. In terms of the convergence rate, whether it is $\sigma$-convergence or $\beta$-convergence, the order was western > central > global > eastern. Therefore, at both the national and regional levels, the regions with a laggard development level of NFM were catching up with the advanced regions at a faster speed, and the development gap would continue to narrow in the future and eventually reach convergence. After considering heterogeneous conditions such as technological inno-

vation, economic development, and education level, the convergence rate was significantly improved, thus indicating that regional socio-economic characteristics can accelerate the convergence rate of NFM synchronous development and shorten the convergence period.

*5.2. Policy Recommendations*

The NFM synchronous development strategy is an important decision made by the Chinese government in the new era. It is an inevitable choice to promote the sustainable and high-quality development of the social economy; it also displays an inherent desire to achieve Chinese modernization. Based on the current situation and existing problems of China's NFM synchronous development, we propose the following policy recommendations:

(1) Improve the development quality of new urbanization and industrialization and create an environment for the rapid development of informatization and agricultural modernization. On one hand, China's industrialization is about to enter the late stage of development, and subsequent development should pay more attention to quality. On the other hand, informationization and agricultural modernization are in the early stages and continue to have great potential; hence, the government should create an environment for their rapid development. First, the government should increase investment in urban health care, elderly care, public transport, and other public services and place social urbanization and ecological urbanization at the forefront. We should establish the concept of ecological city development, expand green space, actively create garden cities, and organically combine the natural environment with urban construction. Second, advanced manufacturing should become the main direction for new industrialization. By promoting the deep integration of the internet, big data, artificial intelligence, and traditional industries, we will cultivate new growth points and form new driving forces in the fields of innovation leadership, green and low-carbon, the sharing economy, modern supply chains, and human capital services. Third, local governments should take measures in accordance with local conditions, encourage the transfer of land in areas suitable for large-scale mechanized agricultural production, encourage the development of agriculture with local characteristics, and increase the diversified supply of agricultural products and farmers' incomes. Fourth, the government can consider increasing private capital investment in information infrastructure to break monopolies and reduce information service costs;

(2) Adapt measures to local conditions and take corresponding measures to promote the synchronous development of NFM. First, we should deepen the integration of industries and cities. Local governments should, according to their own resource endowments and natural conditions, accurately locate industrial development, optimize industrial spatial distribution, and make full use of characteristic industries. Second, urban and rural developments should be coordinated. China should gradually abolish the current household registration system and establish a population management system in which urban and rural residents can move freely. Third, it should unleash the optimization and integration functions of informatization to promote the coordinated development of industry and agriculture. We should focus on developing agriculture with local characteristics, promote the integration of agricultural production and marketing, extend the agricultural industrial chain, and encourage internet companies to establish agricultural service platforms that link production and marketing. Fourth, it should improve the level of agricultural equipment and innovation capacity, and develop standardized, high-yield, and high-efficiency cultivation models.

(3) Strengthen support for the central and western regions and promote a regional balance in the synchronous development of NFM. In view of regional imbalances and coordination problems, policies and funds should be appropriately tilted towards the central and western regions, especially the western regions where NFM development has lagged behind for a long time such as Yunnan, Guizhou, Gansu, and Qinghai. For example, we can consider establishing free trade zones with countries bordering western China to build a new highland for China's opening up. Second, the central and western governments should further improve the market system and deploy talent attraction measures to

stimulate the enthusiasm of various market players. Finally, regional exchanges should be strengthened, and mutually beneficial cooperation mechanisms should be established. Different regions have different resource environments and comparative advantages, which lay the foundation for the regional division of labor and cooperation. Local governments should establish high-level coordination mechanisms, such as regional forums or mayors' joint meetings, to coordinate the major issues of regional cooperation.

(4) Make full use of the synchronous development of NFM and effectively respond to sudden public health emergencies in society. At the end of 2019, COVID-19 broke out in China and spread to the world, lasting three years. High frequency and complex population movements have become a bottleneck restricting urban emergency management and control. Based on digital information technologies, such as big data and the Internet of Things, the Chinese government conducted real-time monitoring, trend analysis, and judgment on urban epidemic risks to achieve effective and accurate epidemic prevention and control. Second, despite the huge impact of COVID-19 on Chinese industry, relying on the industrial internet, cloud computing, and industrial robots, China's industry continues to transform and upgrade, achieving intelligent and high-quality industrial development. Finally, due to the weak integration of agricultural informatization in China, information technologies such as "internet+", artificial intelligence, and big data are still in their infancy in rural areas. As a result, epidemic control has triggered disruptions in agricultural supply chains, leading to rising prices in cities and stagnation in industrial production. It can be seen that promoting the synchronous development of NFM can effectively respond to the opportunities and challenges brought about by public health emergencies.

**Author Contributions:** Conceptualization, K.Z. and B.T.S.X.; methodology, Y.L. and X.L.; software, Y.L.; validation, K.Z., B.T.S.X. and Z.M.T.; formal analysis, X.L. and Z.M.T.; investigation, Y.L. and X.L.; resources, K.Z. and B.T.S.X.; data curation, X.L. and Z.M.T.; writing—original draft preparation, Y.L.; writing—review and editing, K.Z. and B.T.S.X.; visualization, X.L.; supervision, K.Z. and B.T.S.X.; project administration, B.T.S.X.; funding acquisition, K.Z. All authors have read and agreed to the published version of the manuscript.

**Funding:** This research was funded by the National Social Science Foundation of China, grant number 20BGJ016; Henan Province Colleges and Universities Young Backbone Teacher Cultivation Program, grant number 2021GGJS061. The APC was funded by the National Social Science Foundation of China, grant number 20BGJ016.

**Institutional Review Board Statement:** Not applicable.

**Informed Consent Statement:** Not applicable.

**Data Availability Statement:** The data presented in this study are available on request from the corresponding author.

**Conflicts of Interest:** The authors declare no conflict of interest.

## Appendix A

**Table A1.** The division of China's eastern, central and western regions.

| Regions | Provinces |
| --- | --- |
| Eastern | Beijing, Shanghai, Guangdong, Jiangsu, Fujian, Zhejiang, Hainan, Shandong, Liaoning, Hebei, Tianjin |
| Central | Henan, Hubei, Anhui, Shanxi, Jiangxi, Hunan, Jilin, Heilongjiang |
| Western | Shaanxi, Sichuan, Chongqing, Gansu, Guizhou, Inner Mongolia, Qinghai, Guangxi, Ningxia, Tibet, Xinjiang, Yunnan |

**Table A2.** Indicators weights.

| Subsystem | Indicators | Weights |
|---|---|---|
| 0.172Urbanization (UR) | UR1: Urbanization rate (%) | 0.172 |
| | UR2: Per capita disposable income of urban residents (RMB/person) | 0.124 |
| | UR3: Urban unemployment rate (%) | 0.285 |
| | UR4: Per capita park green area (square meters) | 0.168 |
| | UR5: Number of public transportation vehicles (per 10,000 people) | 0.112 |
| | UR6: Urban health technicians (per 1000 people) | 0.139 |
| Industrialization (IND) | IND1: Industrialization rate (%) | 0.08 |
| | IND2: Industrial employment rate (%) | 0.121 |
| | IND3: Industrial labor productivity (%) | 0.175 |
| | IND4: Per capita industrial R&D expenditure (RMB) | 0.53 |
| | IND5: Utilization rate of industrial solid waste (%) | 0.094 |
| Informatization (INF) | INF1: Internet penetration (%) | 0.031 |
| | INF2: Mobile phone penetration (%) | 0.032 |
| | INF3: Per capita mobile switch capacity (ten thousand households) | 0.29 |
| | INF4: Software industry income (ten thousand yuan) | 0.069 |
| | INF5: Number of domain names | 0.349 |
| | INF6: Number of webpages | 0.231 |
| Agricultural modernization (AG) | AG1: Disposable income of rural residents (per capita) | 0.095 |
| | AG2: Electricity consumption of rural residents (per capita) | 0.561 |
| | AG3: Rural mechanical power (per capita) | 0.092 |
| | AG4: Rural effective irrigation area (per capita) | 0.201 |
| | AG5: Number of rural health technicians (per 1000 people) | 0.051 |

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
