# Peer review of "Dynamic Pattern and Evolution Trend of the New Four Modernizations Synchronous Development in China: An Analysis Based on Panel Data from 31 Provinces"

_sustainability, doi:10.3390/su15086745_

Round 1
Reviewer 1 Report
The study of Yang Li et al. represents a valid attempt to scientifically analyze the evolution trend of the New Four Modernizations-NFM synchronous development in China, regarding new urbanization, industrialization, informatization and agricultural modernization. In China, with its vast territory, population and complex geomorphic features, still exists the problem of unbalanced development, also generated by the first program called "Four Modernizations". The topic is relevant and appropriate for this journal.
In general, the work is well organized. The introduction (lines 35-133) is exhaustive and clear. It well explains the state of the art, with over 60 useful and up-to-date references.
The methodological premises and sources that were used by the authors are clearly indicated. The methodology adopted is consistent with the recent literature and with this topic. Various graphs and thematic maps help to follow the discussion. All the calculations seem correct.
Conclusions (lines 731-776) are well argued and provide a great basis for further research. The research is concretely addressed to local governments, to whom some recommendations are addressed (lines 777-835). This is very important!
To improve the paper, it is suggested to include a little reflection on the effects of the pandemic period (2020-2022) on NFM. A punctuation check in the bibliography is also recommended: place always a comma after the surname.
Reviewer 2 Report
1) The title highlighted and begins with spatio-temporal – unfortunately, this study used very little application associated with spatio-temporal. The analysis that applied spatio-temporal is just very brief which is only to display province based on index– and not analyse further the data using other spatio-temporal analysis -such as hotspot, coldspot, trajectory moving, connection between data/historical data. It seems the title need to be reword seems very little use of spatio-temporal concept. This paper is only missing the validation of the output.
2) The abstract is well written start with the intro, purpose, methodology, findings, but missing the problem that this paper attempts to address.
3) In section 3.1 – you describe the indicator and the source. But not clear How you insert the indicators into a model - how far you can confirm the accuracy of the model?
4) In section 3.1.2 there are 3 original data used in this study. For the indicators you mentioned in 3.1, how you extract the data from original source. A snapshot of how the data/field look likes might help to understand if reader plan to replicate your work. For missing data, you run linear interpolation- which data/field are missing- how you confirm the accuracy?
5) Missing in methodology is how you create the spatio-temporal map. Which field involved, how you fit the result from the model and display into the map. From the map, you present the output in provinces- were all the data/field in the original source in province? Please provide the explanation regarding this.
6) Figure 2 – no explanation or in text that referring to this figure.
7) In 4.2.2 - there is no explanation about Panel A and Panel B – if panel referring to people/expert – briefly describe the demographic profile? how many, in which domain/expert?
8) I checked- paper [77] in the reference is not mentioned in the text. Please carefully check and do not include unnecessary paper.
I check the similarity of this paper with others is 31% - where it could be reduced.
Reviewer 3 Report
I am inclined to reject this work. The main points of my negative recommendation include:
(1): As the authors said, the key point is the so-called “synchronization of the 4 modernizations” in China’s recent development. In fact, the authors used a simple algebraic average of the 4 indices, each corresponds to one modernization, to measure the comprehensive development, or the synchronized modernization. I am rather puzzled by the rationality of the proposed measure. Clearly due to geography, different region has different advantages, it seems “synchronized development” is not quite meaningful. For example, Henan province is more suitable for agriculture, Jiangsu province is more suitable for industry, why is the average of the 4 indices meaningful to gauge the developmental status of a region?
(2): The 3 categories of regions are largely affected by a number of big cities, for example, Beijing, Shanghai, disproportionally affects the measured indices in its located region, what are the logic to use such indices to measure whole ”regions”? For Beijing and Shanghai as well as other big cities, it seems to me it is meaningless to measure their agricultural moderation degrees.
(3): In this work, the 3 regions are in fact not defined in the geographical sense, but by their economic development level. For example, Beijing is located much in the north, close to Inner Mongolia, but categorized as “ Eastern”. Considering the “synchronized index” is the average of the 4 different moderation indices, and for better economically better developed provinces, their first 3 indices, UR, IND, and INF, must higher than less developed ones by the current 3-region definition, it seems to me the obtained results in the current work is a foregone conclusion by the regions’ definition;
(4): From the figures, the development curves change linearly in the last 10 years, are such results meaningful? I suppose different region has different development rate. Is it due to the “ linear interpolation” for the missing data?
(5): It seems “2.1 Theoretical basis” is only loosely related to the present work;
Besides, in Eq(4), Eq(5), summation index is not correct; Line 412, “ Equality” should be Quartile
All in all, the current work at its present form is not fit for publication;
Round 2
Reviewer 2 Report
Thank you for well written revised manuscript with a good explanation to clarify my doubts.
Reviewer 3 Report
Although the authors replies do not convince me of the siginificance of the their work , they indeed provide sufficient explanations. I would support its acceptance for publication in the journal.